# Optimizing Anaerobic Digestion at Ambient Temperatures: Energy Efficiency and Cost Reduction Potential in Panama

Euclides Deago [1,2,3], Marian Ramírez [1], Kleveer Espino [1], Daniel Nieto [1], Maudi Barragán [1], Max García [1,4] and Jessica Guevara-Cedeño [1,3,5,*]

1   Research Group Biosólidos Biosolids: Energy and Sustainability, Centro de Investigaciones Hidráulicas e Hidrotécnicas (CIHH), Universidad Tecnológica de Panamá, Panama City 0819-07289, Panama; euclides.deago@utp.ac.pa (E.D.); marian.ramirez@utp.ac.pa (M.R.); kleveer.espino@utp.ac.pa (K.E.); daniel.nieto@utp.ac.pa (D.N.); maudi.barragan@utp.ac.pa (M.B.); magarcia@minsa.gob.pa (M.G.)
2   Sistema Nacional de Investigación (SNI), Panama City 0816-02852, Panama
3   Centro de Estudios Multidisciplinarios en Ciencias, Ingeniería y Tecnología (CEMCIT-AIP), Panama City 0819-07289, Panama
4   Operation and Maintenance, Programa de Saneamiento de Panamá, Reparto Nuevo Panama, Ave. La Fontana, Panama City 0832-02345, Panama
5   Faculty of Electrical Engineering, Universidad Tecnológica de Panamá, Panama City 0819-07289, Panama
*   Correspondence: jessica.guevara@utp.ac.pa; Tel.: +507-560-3048

**Abstract:** Anaerobic digestion (AD) is usually carried out at mesophilic temperatures (25–45 °C) in most countries, whether in temperate or tropical climates, which results in the need for heat injection and consequently increases costs. In this regard, batch AD tests were conducted at 25, 28, and 35 °C, with 25 °C being the lowest ambient temperature in Panama, using thickened secondary sludge (TSS) and digested secondary sludge (DSS) from the Juan Diaz wastewater treatment plant (WWTP) to determine the Biochemical Methane Potential (BMP). The AD study generated maximum mean BMP values of 163 mL $CH_4$/g VS for DSS and 289.72 mL $CH_4$/g VS for codigestion at 25 °C. The BMP value of DSS at 25 °C showed that it can still be used for energy generation, using the lowest ambient temperature recorded in Panama City. Likewise, trials at 25 °C showed a 43.48% reduction in the electrical energy produced compared to that generated at 38 °C in WWTP. This results in a reduction in energy, as the use of heat could be omitted and the energy costs required for the process are covered. In this regard, the novelty of this work lies in its investigation of anaerobic digestion at ambient temperatures, which represents a departure from conventional practices that typically require higher temperatures. By exploring the feasibility of anaerobic digestion within the temperature range of 25–35 °C, this study offers a novel approach to optimizing energy efficiency and reducing costs associated with elevated temperatures.

**Keywords:** anaerobic digestion; biochemical methane; codigestion; kinetic model; bioenergy





## 1. Introduction

Wastewater treatment plants (WWTP) are responsible for purifying wastewater from the population. They should not be seen as a burden on the government or society but rather as an opportunity to recover resources in the form of energy, water, biosolids, and nutrients [1]. In the case of bioenergy, particularly the use of biogas as a renewable source, it plays a crucial role in decarbonization and improvements to climate change. The utilization of organic waste through biogas production not only offers a sustainable energy source but also reduces greenhouse gas emissions by capturing and utilizing methane generated during anaerobic decomposition. Studies such as Sialve et al. [2] have highlighted the relevance of bioenergy and biogas in the transition towards a cleaner energy system and the mitigation of climate change. The reuse of organic sludge also reduces health care costs as a result of proper waste treatment and reductions in energy consumption through the use of biogas, which partially replaces fossil fuels [3].

Worldwide, one of the most important costs in wastewater treatment is electricity, as previous studies have shown that it accounts for 60% of operating costs. However, a WWTP with a biogas recovery system consumes 40% less energy than a WWTP without such a system [4]. The biogas obtained from the anaerobic digestion is often used to heat the digesters and maintain their temperature, requiring large operating costs, mainly in countries with four seasons [5,6]. In the Latin American region, treatment plants struggle to be self-financing since energy costs correspond to 30–40% of operating costs, which has direct implications for the sustainability of the service [1]. Given the need to optimize processes and look for energy-saving alternatives, there are studies where anaerobic digestions are tested at different temperatures and their biogas production potential is evaluated with the objective of evaluating if an acceptable yield is possible at lower temperatures or if a pre-treatment is necessary to compensate for the decrease in biogas production [7].

In the case of Colombia, the country has focused on digesters for rural areas where they are operated at psychrophilic temperatures with biogas on-site [8]. Despite advances in innovation and development, there are still deficiencies in policies that regulate the biogas generation chain, and these improvements would help to develop an energy production technology for the country [8]. The technology of tubular digesters has been expanded in Costa Rica, Nicaragua, Ecuador, and Honduras. In Mexico, research in the field of energy production through biogas began in 1990, and the study of energy recovery through methane has been extensively studied [9]. An example of this is studied by Alcaraz et al. [10], where a study at low temperatures (17 and 19 °C) records energy yields that would supply the needs of up to 42 households in the Toluca area.

The study of anaerobic digestion in Panama is something new and has been carried out for a few years [11–13]. This has been the product of a reorganization of sanitary sewage with the construction of sanitary networks, collector lines, the Juan Diaz Wastewater Treatment Plant (WWTP), and the implementation of the Panama Bay Sanitation Program. These measures have been put in place with the aim of recovering sanitary and environmental conditions, reducing pollution in the urban rivers of Panama City, and improving the sanitation of the Bay of Panama [14]. However, it is a challenge for Panama to make sanitation advancements since once organic sludge is generated in the Juan Díaz WWTP, it is dehydrated and stabilized and subsequently disposed of in the landfill at a daily rate of approximately 80 tons per day [15].

This situation of sludge disposal is in line with that mentioned by Kiselev et al. [16], where the use of sewage sludge remains a global problem in need of an adequate solution. In the case of Panama, prior to the disposal of the stabilized sludge, the Juan Díaz WWTP achieves its partial use through the anaerobic digestion (AD) of the sludge, generating biogas that allows an approximate electric cogeneration of 20% of the demand required by the WWTP [11,17]. This AD is performed at a temperature of 38 °C, which is conventional for sludge treatment. This process is a small-scale application of the circular economy concept, where organic sludge is used as a raw material for biogas generation, which in turn generates electricity.

In this sense, Panama has a regulatory framework that encourages electricity generation using new and clean technologies through the implementation of private electric generation plants of up to 500 kW capacity [18]. There are also national policies, such as the National Energy Plan 2020–2030 (PEN), in which the decarbonization of the country's energy matrix and its diversification are included in the guidelines [19]. In addition, until 2030, Panamá has the opportunity to implement the Sustainable Development Goals (SDGs), where progress would be made in the SDG related to energy through the generation of electricity in wastewater treatment plants using the biogas produced, as well as the SDG13 treating the water residuals [20]. These regulations are in addition to the Law on Integral Management of Solid Waste, which promotes the safe use of hazardous waste, either in the same industry or as a raw material or alternative fuel [21].

With all these regulatory frameworks and objectives in sanitation issues, a bibliographic search was made about other possible uses for digested secondary sludge (DSS),

so this waste can be reduced further and converted into a resource. Digested sludge is a by-product of anaerobic digestion in wastewater treatment plants. These sludges, after being digested, have high concentrations of nutrients such as C, N, P, and K [22,23]. In this respect, they have the potential to be exploited in agriculture [23]. Also, Digested sewage sludge can generate heat and electricity through incineration [22]. In this sense, it is known that Anaerobic Digestion (AD) of organic sludge for energy is beneficial to reduce global warming and even more efficient than treatments such as composting and incineration [24].

The reason there is still organic material available in the already digested anaerobic sludge is the fact that the retention times managed in most WWTPs range from 20 days, where shorter periods can reduce the efficiency of the digestion and result in low biogas yields, resulting in digested sewage sludge that still has usable potential [4]. Cao and Pawlowski [25] conclude that anaerobic digestion alone is not capable of fully recovering energy from residual sludge; therefore, the digested sludge continues to be energetically beneficial. The anaerobic biomass present in the digested sludge makes it suitable for use as inoculum [25]. There are several studies where it has been used as an inoculum in the co-digestion of substrates such as grease traps [26], wood chips [27], and organic waste [28].

The objective of this research is to evaluate and optimize biogas production through anaerobic digestion at ambient temperature in a wastewater treatment plant, focusing on the use of digested secondary sludge (DSS) as a valuable feedstock. The research aims to enhance biogas production efficiency, explore innovative technologies, and contribute to the advancement of sustainable wastewater management practices.

The novelty of this work lies in its investigation of anaerobic digestion at ambient temperatures, which represents a departure from conventional practices that typically require higher temperatures. By exploring the feasibility of anaerobic digestion within the temperature range of 25–35 °C, this study offers a novel approach to optimizing energy efficiency and reducing costs associated with elevated temperatures. Furthermore, the research contributes to the promotion of cleaner energy sources by assessing the potential of utilizing sewage sludge as a renewable energy resource. The focus on energy cost minimization and the reduction of greenhouse gas emissions, specifically through the capture and utilization of methane, presents novel insights for sustainable waste management practices in Panama.

## 2. Materials and Methods

### 2.1. Sludge Sampling

The samples were taken at the Juan Diaz Treatment Plant, located in the township of Juan Diaz in the district of Panama (9°0′54.8″ N 79°26′43.3″ E). The Juan Díaz WWTP treats an average wastewater flow of 2.8 $m^3$/s in module 1, where sampling was carried out. Sampling was carried out monthly for a period of five months, from November to March. There were two types of sludge used in this research: (i) thickened secondary sludge (TSS), which was purged from the secondary sedimentator of the Juan Díaz WWTP and then underwent a thickening process where polymers were added; and (ii) digested secondary sludge (DSS), which underwent anaerobic digestion at a mesophilic temperature of 38 °C inside the WWTP [17]. The samples were preserved at a temperature of 4 °C and then transported to the laboratory following preservation criteria [29,30].

### 2.2. Analytical Methods

The sludge samples were tested for their physicochemical properties and their compliance with the Technical Regulations for the Use and Disposal of Sludge [31]. The methodologies followed to define the parameters were pH and conductivity (multiparameter, Hach, Loveland, CO, USA), total solids and volatile solids (VS) with Standard Methods; Chemical Oxygen Demand (COD)[32], alkalinity, and Volatile Fatty Acids were analyzed using DR6000 UV-VIS, Hach, USA. The elemental analysis of sludge was performed following the methods detailed in AOAC (2005) [33,34].

*2.3. Anaerobic Digestivity Test*

Digestibility tests were carried out in $n = 3$ for each temperature (25, 28, and 35 °C) and for each type of sample. The reactors were glass vessels with a total volume of 500 mL. A total of 54 reactors were used in the study. Ambient conditions in the laboratory were maintained at 21 °C. The digestibility test was performed at a constant controlled temperature, although the WTW incubator model TS 608-G/2i used handled a range of temperatures between ±0.1 °C.

According to Owen [35], in most cases, anaerobic digestion trial incubation is required to last for 30 days for complete degradation. A sludge volume (TSS and DSS) of 300 mL was used for the trials. The pH range during the assay was between 6.5 and 7.5 using $K_2HPO_4$ (2 g/L) and $KH_2PO_4$ (2 g/L) [36].

In this study of the anaerobic digestivity of organic sludge, the Oxitop® System was used [37]. The System works with the manometric method and consists of recording deltas of pressures generated internally in batch reactors by the metabolic activity of microorganisms.

One aspect of the Oxitop® System is the ability to incorporate sodium hydroxide (NaOH) beads internally into reactors to capture carbon dioxide generated in anaerobic digestion, known as the $CO_2$ trap, which allows head sensors to record the net pressures produced by methane gas. According to Souto et al. [36], the use of sodium hydroxide beads captures $CO_2$ from the bottle, leaving only methane gas. The system also has a magnetic base that shakes the contents of the batch reactors in both directions to maintain their homogeneity. This methodology has been used successfully in similar research [36,37].

2.3.1. Codigestion

Anaerobic codigestion was carried out with DSS as inoculum and TSS as substrate, retaining the same useful volume of 300 mL in a 500 mL reactor used in monodigestion tests. An inoculum/substrate (I/S) ratio of 2:1 was used, as it is proven to have better performance in anaerobic codigestion [38]. Inoculum/substrate ratios in codigestion were based on a ratio of Volatile Solids [39]. It is recommended that the I/S ratio be greater than 2 to prevent acidification, especially in tests with rapidly biodegradable substrates [40]. In Figure 1, we present an anaerobic digestion scheme in WWTP and proposed anaerobic co-digestions.

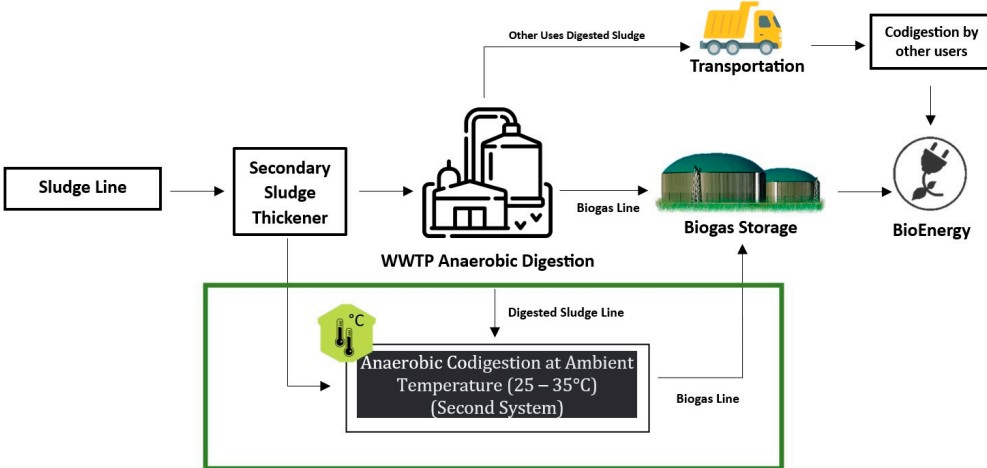

**Figure 1.** Anaerobic digestion scheme in WWTP and proposed anaerobic co-digestions.

2.3.2. Calculation of the Biochemical Potential of Methane

The biochemical potential of methane (BMP) is a test that gives a measure of the biodegradability of a substrate and is determined by monitoring the production of methane accumulated in a batch reactor operating under anaerobic conditions in a chemically controlled environment [35]. This experimental BMP is obtained by means of a series

of equations where the pressure values recorded by the Oxitop® System are introduced, corresponding to the methane produced. This method has been used in research on the anaerobic digestivity of organic sludge and other biodegradable matrices with satisfactory results [12,41–43].

### 2.3.3. Kinetic Model of Methane Prediction

There are many models used in various investigations to simulate the production of biogas resulting from biological degradation through anaerobic digestion [44]; however, the most widely used model that best fits digestivity test data is the Modified Gompertz [45,46]. This is an experimental-type model of nonlinear regression that describes the growth of bacteria in batch assays of anaerobic digestivity [47]. The modified Gompertz equation (Equation (1)) was generated with the Microsoft Excel SOLVER add-in program to perform the simulations.

$$BMP_i = P_{max} * exp\left[-exp\left(\frac{R_{max} * Exp(1)}{P_{max}}\right)(\lambda - t) + 1\right] \tag{1}$$

where $BMP_i$ is the biochemical potential of accumulated methane (mL $CH_4$/g VS); $P_{max}$ corresponds to the maximum value of BMP recorded at the end of the test (mL $CH_4$/g VS); $R_{max}$ is the maximum methane production rate (mL $CH_4$/g VS/day); $\lambda$ is the lag phase (day); and $t$ is trial time in days.

### 2.3.4. Biodegradability Kinetics of Sludge

To estimate the degradability rate, a first-order model (Equation (2)) was used, where $k_H$ is the degradability rate (days$^{-1}$) and $BMP_{max}$ is the methane potential of the substrate (mL $CH_4$/g VS).

$$BMP_i = BMP_{max}\left(1 - exp^{(-k_H\,t)}\right) \tag{2}$$

With this model, the biodegradation of macromolecules was estimated, since it establishes hydrolysis as a limiting step, which has been proposed by several authors for digestibility studies [44,47].

### 2.4. Energy Estimation

The Combined Heat and Power system (CHP) is a combination of technologies to produce electricity and thermal energy from various fuels. These systems allow the use of renewable fuels such as biogas or biomass for cogeneration [48]. The Juan Diaz WWTP uses the biogas flow as fuel in a CHP energy system for electricity and thermal production. For energy estimation, an adapted methodology has been used from Szaja et al. [49,50]. For the calculation of the daily methane production (Equation (3)), $Q_{CH4}$ (m$^3$ $CH_4$ d$^{-1}$) is given by $Y_m$ (m$^3$ $CH_4$ kg$^{-1}$ VS$_{add}$) is the methane yield, and $L_{VS}$ (kg VS d$^{-1}$) is the volatile solids (VS) load in the feedstock.

$$Q_{CH4} = Y_m \times L_{VS} \tag{3}$$

The theoretical amount of thermal energy obtained from the combustion of methane is given by $Q_t$ (MJ d$^{-1}$) and $Q_i$—the heating value of methane (35.8 MJ m$^{-3}$); using Equation (4),

$$Q_t = Q_{CH4} \times Q_i \tag{4}$$

There are some variables, used for the estimation, which are described here below: $V_{os}$ —feedstock flow rate, m$^3$ d$^{-1}$; $C_{SS}$ —the specific heat of sewage sludge, kJ m$^{-3}$ K$^{-1}$ (4200 kJ m$^{-3}$ K$^{-1}$). The difference between the temperature of anaerobic digestion ($T$), the ambient temperature ($T_{air}$), and the temperature of feedstock ($T_{feed}$) is K; $U$ —the heat loss coefficient by permeation through the walls of the digester, (4.0 kJ m$^2$ h$^{-1}$ K$^{-1}$); $D$ —the

diameter of the cylindrical part of the digester, m. The surface of the digester walls, $F$ ($m^2$), was calculated with Equation (5).

$$F = 3.17 \times D^2 \tag{5}$$

It is necessary to calculate the required thermal energy to heat the digester and to cover the heat loss, where Equation (6) is the thermal energy required for heating the feedstock $Q_1$, kJ $d^{-1}$, and Equation (7) is the thermal energy required to cover the heat loss through the walls of the digester $Q_2$, kJ $d^{-1}$.

$$Q_1 = V_{os} \times (T - T_{feed}) \times C_{SS} \tag{6}$$

$$Q_2 = 24 \times (T - T_{air}) \times U \times F \tag{7}$$

To know the thermal energy demand required for the digester (kJ $d^{-1}$), we used Equation 8 with a margin factor of 1.1.

$$Q_c = 1.1 \times (Q_1 + Q_2) \tag{8}$$

The daily energy production $E_d$, kWh $d^{-1}$, is given by using Equation (9); also, Equation (10), the $\eta_{t/e}$ —the thermal/electric efficiency of the CHP engine ($\eta_t = 0.43$, $\eta_e = 0.38$).

$$E_d = Q_{CH4} \times Q_{ie} \tag{9}$$

$$N_{t/e} = (Q_{CH4}/24) \times Q_{ie} \times \eta_{t/e} \tag{10}$$

## 3. Results and Discussion

### 3.1. Physicochemical Characterization

The average of the results obtained during the five months of sampling the residual sludge of the Juan Díaz WWTP are shown in Table 1.

**Table 1.** Characterization of the inoculum and substrate.

| Parameters | TSS | DSS | I/S = 2/1 |
|---|---|---|---|
| pH | $7.00 \pm 0.20$ | $7.87 \pm 0.31$ | 7.41 |
| COD (mg/L) | $2406 \pm 808.84$ | $2669.50 \pm 330.07$ | $1903 \pm 110.15$ |
| Conductivity (µs/cm) | $519.4 \pm 240.17$ | $1498.2 \pm 239.40$ | - |
| Volatile Solids (%) | $80.66 \pm 0.14$ | $76.72 \pm 0.18$ | $78.73 \pm 0.42$ |
| Total Solids (% vs.) | $0.66 \pm 0.02$ | $0.41 \pm 0.02$ | $0.45 \pm 0.006$ |
| Alkalinity (mg/L) | $716 \pm 76.37$ | $4116.66 \pm 270.06$ | $2663 \pm 219.39$ |
| Volatile Fatty Acids (mg/L) | 1160 | 1850 | 1447 |
| C (%) | 2.00 | 1.70 | 38.28 |
| N (%) | 0.38 | 0.47 | 6.25 |
| H (%) | - | - | 5.20 |
| O (%) | - | - | 16.30 |
| C/N | 5.26 | 3.61 | 6.12 |

Notes: COD—Chemical Oxygen Demand; C—Carbon; H—Hydrogen; N—Nitrogen; O—Oxygen; C/N—Carbon/Nitrogen Ratio.

TSS and DSS presented pH values between 7.00 and 8.18, which allow methanogenic bacteria to survive and organic matter degradation to occur [51]. The total solids obtained for TSS and DSS, 0.66 and 0.41, respectively, are in the range obtained by Mitraka et al. [52], where for activated sludge, they were between 0.4 and 1.2. The inoculum, substrate, and mixture presented values of volatile fatty acids less than 2000 mg/L $CH_3COOH$; this

indicates a non-toxic environment for methanogenic bacteria [53]. The C/N ratio was low compared to the optimal range, which should be 20–30 for anaerobic digestion [54].

### 3.2. Digestivity Assays and Biochemical Methane Potential

The experimental results show a variation in the generation of biogas, or BMP, in the anaerobic digestion of the sampled sludge. This test was developed in an endogenous environment, and the results correspond to tests performed on samples collected monthly for five months. Batch tests were carried out at 25, 28, and 35 °C to establish a possible operational advantage by eliminating the use of heat to keep reactors at 38 °C, as is currently the case [52]. Panama has a natural advantage due to its tropical climatic conditions, with maximum average temperatures of 32.4 °C to 35.1 °C [55]. In the case of the present work, the temperature of 25 °C was taken as a minimum reference temperature in the mesophilic range, since throughout the day the temperature increases until it is around the maximum average described above.

The results of the monodigestion at 25 °C, TSS, and DSS trials yielded maximum values of 130.39 mL $CH_4$/g VS and 163.09 mL $CH_4$/g VS at a retention time of 30 days, respectively. The methane yield obtained by DSS shows what Lanko et al. [56] indicate: the digested sludge is the best to be used as fuel because it has organic matter that was not transformed into biogas in the previous anaerobic digestion. Due to the values obtained in digestion, it was decided to make a codigestion between thickened secondary sludge and digested secondary sludge at a temperature of 25 °C. Figure 2 shows the biochemical methane potential of monodigestion and codigestion at different temperatures.

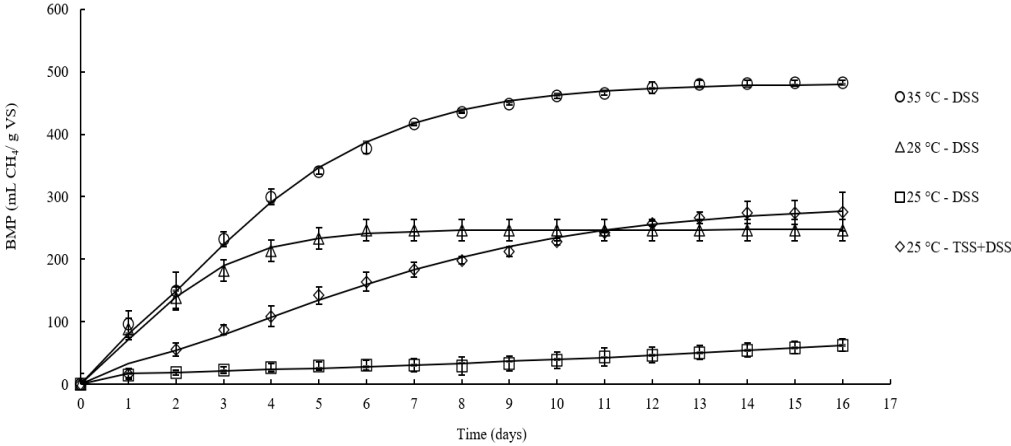

**Figure 2.** Biochemical potential of methane at different temperatures. A continuous line represents a good fit of the modified Gompertz model to the experimental values.

An improvement in BMP performance was observed in the codigestion test at 25 °C, recording the maximum stabilization value at 289.72 mL $CH_4$/g VS (Figure 2). The stabilization occurred at day 6, where a more accelerated methane production could be verified compared to the DSS monodigestion at 25 °C, where even at day 16, a noticeable stabilization still did not occur. Also, DSS were tested at higher temperatures, such as 28 °C and 35 °C, obtaining BMP values of 246.91 mL $CH_4$/g VS and 482.50 mL $CH_4$/g VS, respectively. With the exception of DSS at 25 °C, the monodigestions and codigestions began to stabilize after 10 days; this indicates the advantage of an acclimatized sludge, where anaerobic digestion processes are favored [57]. The results obtained for digested sludge (DSS) motivated the use of this sludge for monodigestion at two higher temperatures, 28 °C and 35 °C, lower than the 38 °C used by the Juan Díaz WWTP and still in the range of ambient temperatures in Panama City [55].

When comparing the results obtained from the mono and codigestion trials of TSS and DSS with data reported from recent research on anaerobic digestions under mesophilic conditions (20–45 °C), it is observed that they are within the expected ranges (Table 2). In

addition, BMP assays are commonly expressed in different units, which makes comparison between studies difficult when making a bibliographic review, as shown in Table 2.

**Table 2.** Comparison of the Biochemical Methane Potential obtained in this study with respect to the data reported in the literature.

| Type of Sludge | BMP (mL $CH_4$/g VS) | Temperature (°C) | Reference |
|:---:|:---:|:---:|:---:|
| LP + LS | 65<br>129 | 20<br>40 | [51] |
| LP + LS | 243<br>271<br>274 | 32<br>34.5<br>37.5 | [58] |
| LS | 143 | 37.5 | [59] |
| LP | 220 | 37 | [60] |
| LP | 188 | 37 | [61] |
| LP + LS | 146 | 37 | |
| LS | 229–233 | 35 | [62] |
| LS | 190 | 35 | [28,63] |
| TSS | 130.39 | 25 | This Study |
| DSS | 163.09 | 25 | This Study |
| TSS + DSS | 289.72 | 25 | This Study |
| DSS | 246.91 | 28 | This Study |
| DSS | 482.50 | 35 | This Study |

Notes: LS—Secondary Sludge; LP—Primary Sludge; TSS—Thickened Secondary Sludge; DSS—Digested Secondary Sludge.

There are factors that could affect the BMP values obtained between this trial and those reported in the literature [64]; for example, in our study, the agitation was constant, and in a similar study conducted by Julio [28], the agitation was manual. This condition can affect mass transfer between substrates and microorganisms [65]. In addition, in the Julio study, the VS was 67.17%, while in this study, the VS was 41.17%, which indicates that with more organic content and bacterial biomass, its production would be higher [38].

It is important to note that, although heating the bioreactors could increase biogas production, the continuity of digestion in smaller reactors also has its advantages. The hydraulic retention time (HRT) of anaerobic digestion, which is generally around 16 days, may be suitable for the use of DSS in codigestion. This means that the strength of the data generated by continuing digestion in smaller reactors can be exploited, rather than relying solely on a large and expensive bioreactor [66].

### 3.3. Digestivity Kinetics

To evaluate the kinetics of digestivity batch assays, the Modified Gompertz model yields interesting parameters to understand how the microbiological population behaves. There is the phase of lethargy or acclimatization ($\lambda$) and the maximum methane production rate ($R_{max}$). In all the digestibility tests carried out, high precision was obtained in the adjustment of the modified Gompertz model with respect to the experimental data (Figure 2). All values of $R^2$ were above 0.995 (Table 3).

**Table 3.** Results of the Modified Gompertz Model of tests run at 25, 28, and 35 °C.

| Type | Temperature (°C) | $R_{max}$ (mL CH$_4$/g VS/day) | $\lambda$ | R$^2$ |
|---|---|---|---|---|
| TSS | 25 | 4.70 | 1.53 | 0.9985 |
| DSS | 25 | 4.52 | 0.46 | 0.9827 |
| DSS + TSS | 25 | 27.70 | 0.13 | 0.9956 |
| DSS | 28 | 70.85 | 0 | 0.9951 |
| DSS | 35 | 74.74 | 0 | 0.9986 |

With respect to the kinetic parameters of the tests with TSS tested (Table 3), a phase of lethargy or acclimatization of 0.46 days can be noted according to Table 3 for a test period of 16 days.

In the case of monthly TSS digestivity assays, the Gompertz equation modified the maximum methane production rate ($R_{max}$), which on average is 4.52 mL CH$_4$/g VS/day, and the duration of the lethality or acclimatization phase ($\lambda$) was 0.46 days (Table 3). As previously indicated, DSS was obtained from anaerobic digestion at a digestion temperature of 38 °C (WWTP) and was then used in the digestivity test at a temperature of 25, 28, and 35 °C. The value of the lethargy or acclimatization phase ($\lambda$) was zero days. This result allows us to infer that the previous digestion in the WWTP helped the acclimatization of anaerobic bacteria. In addition, degradable organic material is available, and for this reason, the lethargy phase is zero, as previously reported [42]. Meanwhile, the model showed that the values of the parameter $R_{max}$ were in the range of 4.0 to 7.0 for TSS and DSS (Table 3). The codigestion at a temperature of 25 °C obtained values of $R_{max}$ of 27.73 mL CH$_4$/g VS and $\lambda$ of zero days. The $R_{max}$ results for DSS at 28 °C and 35 °C were 70.85 and 74.74 mL CH$_4$/g VS, respectively. In this sense, it is known that the higher the temperature, the higher the rate of methane production.

The Gompertz-modified settings for the experimental results of DSS + TSS codigestion reflect a higher performance than TSS and DSS monodigestions, in addition to having a short acclimatization phase time. This means that codigestion at a mesophilic temperature of 25 °C allows for greater biogas production with a shorter acclimatization time and a shorter system stabilization period. These conditions would be favorable in terms of reducing the energy costs associated with the heating of anaerobic digestion reactors, mainly in tropical climate regions where temperatures fluctuate in mesophilic ranges. According to the results shown by Chae et al. [67], it is essential to maintain the digesters at an optimal temperature because the interaction of the bacteria is very sensitive to temperature changes. However, their results showed that a correct adaptation of the mesophilic bacteria can be achieved as long as it is conducted in a very gradual way.

Anderson et al. [58] indicate that while it is true that there is a reduction in biogas production with decreasing temperatures, this could be offset in terms of cost savings due to the issue of warming. Also, it was found that the proportion of methane in the biogas generated was maintained. In addition, although it is known that by increasing the temperature to a thermophilic range (50 °C), a greater volume of methane can be obtained, this happens at a longer hydraulic retention time and acclimatization phase with a decrease in microbial stability, and one must also consider the energy expenditure represented by maintaining reactors at this temperature [51,68].

### 3.4. Biodegradability Kinetics of Sludge

For the degradability rate ($k_H$), the first-order equation was used. Values of 0.04 d$^{-1}$ and 0.02 d$^{-1}$ were obtained for trials with TSS and DSS, respectively. These hydrolysis results are consistent with those seen in the literature [69,70]. For the Codigestion of TSS + DSS at 25 °C, a degradability rate of 0.16 d$^{-1}$ was obtained, which is very positive and indicates that the substrate inoculum mixture had a rapid degradation [71]. For DSS monodigestion at 28 °C and 35 °C, $k_H$ values of 0.47 and 0.25 were obtained.

### 3.5. Energy Estimation for CHP System

It was considered prudent to estimate energy production for the DSS at 25 °C compared to that produced by Juan Diaz WWTP at 38 °C since the DSS performs better in the monodigestion stage. Figure 3 shows the methane productions at 25 and 38 °C and the demands required by the WWTP.

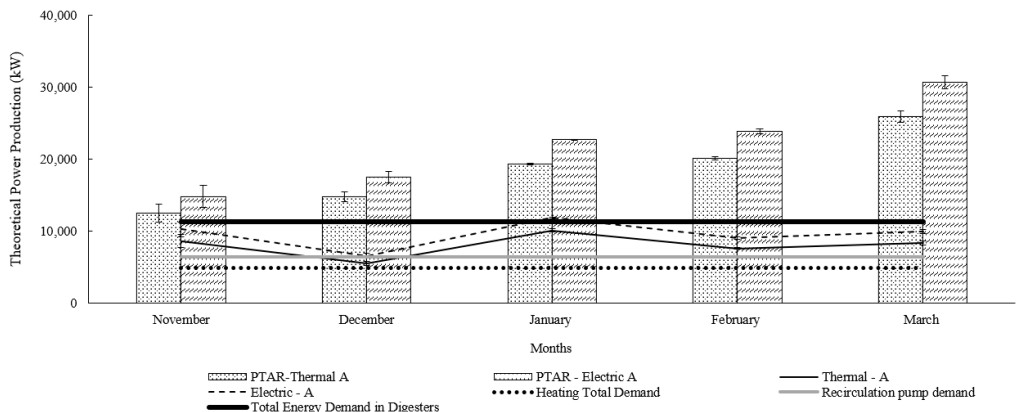

**Figure 3.** Comparison of Energy Estimation for the CHP System of WWTP Juan Díaz and batch estimation at 25 °C.

The results of the study demonstrate that the production of the Juan Diaz wastewater treatment plant (WWTP) at 38 °C is 43.48% higher compared to the batch tests conducted at 25 °C (Figure 3). The differences in electrical and thermal generation were found to be 15.58% and 15.75%, respectively. At an average temperature of 38 °C, the estimated electrical output from the WWTP was 21,912 kW/month, with a thermal energy output of 18,498 kW/month. However, at a temperature of 25 °C, the lowest temperature in the mesophilic range, the batch tests yielded an estimated electrical output of 9528 kW/month and a thermal output of 8028 kW/month. The study also revealed that the total energy required by the digesters for heating and sludge recirculation averaged 11,376 kW/month, with approximately 4896 kW/month needed for the hot water pump. Interestingly, even in the month with the lowest thermal energy production (December) at 25 °C (5580 kW/month), it was found that the minimum thermal energy required to maintain the digesters at an average temperature of 38 °C would still be met. In comparison, there is an average electrical power output per WWTP of 21,912 kW/month, while the output of the 25 °C test was 9528 kW/month. In this case, the energy required per month for recirculation of sludge is 6480 kW/month, so we can note that the needs of digesters would also be met in terms of the energy necessary for their operation. In this sense, we can infer that a production at 25 °C can supply the energy needs required by the plant for heating the digesters. The electrical production given by the plant at 38 °C in the month of lower production was 39.04% more than the demand for heating. According to Moestedt [72], there is evidence that with changes in temperature, in his case from 42 °C to 34 °C, more biomass is managed and there is less methane production. Also, one of the factors that affect the economy of the WWTP is, in part, the energy used in the heating of digesters and the electricity necessary for aeration.

Codigestion is an option that could be tested at different temperatures, based on results from 25 °C. Its methane production was stable for 16 days compared to the retention times processed by the WWTP, which oscillate within 22 days at 38 °C. This case indicates that the mixture of biomasses and the addition of SV to the DSS inoculum, through the addition of a TSS substrate, boosted methane production, allowing a more accelerated production that represents double that produced by a single substrate. Andersson et al. [58] noted that lower reactor temperatures result in lower energy costs. Their study showed that a 2.5 °C decrease in operating temperature represented 13% energy savings; this result was obtained in laboratory-scale tests measuring the effect of temperature on biochemical

methane production. Also, another study, where a comparison was made between different mesophilic temperatures (35 °C and 39 °C), also shows that temperature modification is a feasible option to find optimal biogas production since the additional energy needed for the temperature increase in the reactors is balanced by the amount of extra gas produced at a higher temperature [58].

In the literature, it has been suggested that an economical method for maintaining the temperature in anaerobic digestion processes is by utilizing the excess thermal energy from biogas through a combined heat and power (CHP) method. Deng [7] highlights this approach to optimizing biogas production. However, it is important to note that the excess thermal efficiency achievable through this method is limited, typically ranging from 40% to 45%. Kougias and Angelidaki [73] also support this limitation in their research.

Therefore, in the experimental study conducted, it became crucial to optimize biogas production considering these parameters. The utilization of excess thermal energy from biogas through a combined CHP approach can contribute to increasing the overall energy efficiency of the anaerobic digestion process. By capturing and utilizing the thermal energy generated during the biogas production, the system can minimize energy losses and maximize the utilization of available resources.

It is essential to consider the economic feasibility and practical implementation of the combined CHP method in the context of the specific anaerobic digestion system. Factors such as the efficiency of energy conversion, infrastructure requirements, and operational costs need to be thoroughly assessed to determine the viability and potential benefits of implementing this approach.

## 4. Conclusions

Co-digestion of DSS offers an interesting opportunity to improve biogas production and maximize the utilization of available resources. While DSS itself is a valuable substrate, its use can be further enhanced by considering the addition of thickened sludge as an additional substrate. This strategy could increase the organic and nutrient loads in the anaerobic digestion process, resulting in increased biogas production.

It was found that DSST does not require an acclimatization stage ($\lambda$ = 0 to 0.46), given the digestion carried out in the WWTP Juan Díaz. This is because the active microbial flora was acclimatized, active, and mature enough to quickly initiate anaerobic digestion. This is very favorable for subsequent use in anaerobic codigestion, that is, in the generation of biogas. The retention time in the monodigestion of DSS did not reach a stabilization phase even at 16 days, which compares with reports of monodigestion made in similar studies; in contrast, we can observe that in the codigestion of TSS + DSS, the stabilization phase occurred at 14 days, after an acclimatization phase of 0.13 days following the start of the trial.

While it is true that the ambient temperature of 25 °C showed a low rate of methane production $R_{max}$, BPM data remained in ranges such as those found in studies at 35 °C and 37 °C. It is important to evaluate the energy savings that can occur in countries with hot climates and the savings involved in reducing the heating of reactors and performing codigestions that accelerate the production process to a stabilization phase. The favorable results of BMP at a more critical ambient temperature were demonstrated in Panama (25 °C), which showed a 43.48% reduction in the electrical energy produced compared to that generated at 38 °C in WWTP. Although a reduction in production was observed, the output is still sufficient for the energy requirements of the digesters.

These results allow us to infer that there is great potential for reusing DSS in a circular economy, either internally within the WWTP or by outsourcing it to another user. The use of digested secondary sludge for biogas has considerable additional potential. Optimization of anaerobic digestion, improvements in biogas capture and utilization technology, the implementation of efficient management systems, and continuous research and development can increase this additional potential and promote the sustainable use of secondary sludge as a source of renewable energy and waste management.

**Author Contributions:** Conceptualization, E.D. and M.R.; methodology, K.E., M.R. and J.G.-C.; writing—review and editing, M.R., K.E. and E.D. and J.G.-C.; supervision, E.D. and M.G.; project administration, D.N. and M.B.; funding acquisition, E.D. All authors have read and agreed to the published version of the manuscript.

**Funding:** This research was funded by Secretaria Nacional de Ciencia, Tecnología e Innovación de Panamá (SENACYT), project number IDDSE19-008.

**Data Availability Statement:** Data supporting the reported results can be obtained in Figures 2 and 3, Tables 1 and 3 in the original manuscript. For more details in data presented in this study, are available on request from the corresponding author.

**Acknowledgments:** The authors wish to thank WWTP of Juan Díaz for providing samples and facilitating the execution of the research work. They thank the Universidad Tecnológica de Panamá for providing the facilities for the development of this project, and thank the Secretaría Nacional de Ciencia, Tecnología e Innovación (SENACYT) and Centro de Estudios Multidisciplinarios en Ciencias, Ingeniería y Tecnología AIP (CEMCIT-AIP).

**Conflicts of Interest:** The authors declare no conflict of interest.

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
