# Peer review of "Optimizing Anaerobic Digestion at Ambient Temperatures: Energy Efficiency and Cost Reduction Potential in Panama"

_water, doi:10.3390/w15142653_

Round 1
Author Response
I would like to thank you on behalf of myself and my colleagues for the review of our manuscript.

Reviewer 2 Report
This study aims to investigate the anaerobic digestion of organic sludge at temperatures of 25, 28 and 35 °Ð¡. Generally, the topic is interesting, and the research is professionally carried out and the manuscript is well written. This paper is therefore recommended for publication.
Author Response

(The authors gave the same response as above.)

Reviewer 3 Report
The work shown is interesting but it looks like a simple technical report in its current form. More discussion and explanation are required regarding the results obtained and comparison against literature as well. Model fittings should be verified. Detailed comments are as follows:
1. Some language errors must be corrected:
Line 100: “The reactors are glass” – Keep tense consistency.
Line 190: “for calculate”.
Line 217: “are shown (Table 1).” – correct into “are shown in Table 1”.
Line 245: “Figures 1”.
Lines 278-289: Revise the grammar; the meaning in this paragraph is lost.
Line 354: “to estimate energy production”.
Line 356: “In Figure 2, shows”.
2. Abstract:
The text in lines 30-33 cold be omitted; instead, the novelty of the work could be highlighted.
3. Introduction:
a) Although Introduction is very informative regarding Panama’s case, it does not cite any similar works in the field. So, a literature review is required, which could be further used to compare the results obtained.
b) At the end of Introduction, highlight the novelty of the work.
4. Materials and Methods:
a) Provide the location (coordinates) and the capacity of the Juan Díaz WWTP. Which months specifically did sampling take place?
b) Equation 1: Explain Pmax and provide the units used.
c) Equation 2: Explain kH and provide the units used.
5. Results and Discussion:
a) Figure 1: Correct y-axis title from “PBM” to “BMP” for consistency. Translate the x-axis title into English. Do not use decimal points for y-axis values. Change the scale of x-axis. Provide the main experimental details per line.
b) Figure 1: The authors mention that “A continuous line represents a good fit of the modified Gompertz model to the experimental values”: Are the authors certain about the fact that the lines shown represent the model fitting? Especially in the case of 35oC-DSS, the curve passes exactly from all points changing slope for every two successive points.
c) Use a standard abbreviation for the biochemical potential of methane throughout the text.
d) Figure 2: The error bars are missing in the case of January.
e) The results shown in the paragraph 3.6 are not compared against literature.
f) Present the results obtained in a more scientific way than descriptive.
6. Conclusions:
Provide the objectives of the work at the beginning.
Comments about the language have been provided above.
Author Response

(The authors gave the same response as above.)
